# What Is the Minimum Number of Sutures for Microvascular Anastomosis during Replantation?

**DOI:** 10.3390/jcm12082891

**Published:** 2023-04-15

**Authors:** Hyung-suk Yi, Byeong-seok Kim, Yoon-soo Kim, Jin-hyung Park, Hong-il Kim

**Affiliations:** Department of Plastic and Reconstructive Surgery, Kosin University Gospel Hospital, Kosin University College of Medicine, Busan 49267, Republic of Korea

**Keywords:** reconstructive surgical procedures, microsurgery, suture technique, skin grafting

## Abstract

As vessel diameter decreases, reperfusion after anastomosis becomes more difficult. When a blood vessel is sutured, its inner diameter becomes narrower owing to the thickness of the suture material and the number of sutures. To minimize this, we attempted replantation using a 2-point suture technique. We reviewed cases of arterial anastomosis in vessels with a diameter of less than 0.3 mm during replantation performed over a four-year period. In all cases, close observation was followed by absolute bed rest. If reperfusion was not achieved, a tie-over dressing was applied, and hyperbaric oxygen therapy was administered in the form of a composite graft. Of the 21 replantation cases, 19 were considered successful. Furthermore, the 2-point suture technique was performed in 12 cases, of which 11 survived. When three or four sutures were performed in nine patients, eight of these cases survived. Composite graft conversion was found in three cases in which the 2-point suture technique was used, and two of these cases survived. The survival rate was high in cases where 2-point sutures were used, and there were few cases of conversion to a composite graft. Reducing the number of sutures aids in optimizing reperfusion.

## 1. Introduction

Accurate and successful microvascular anastomosis is essential for free tissue transfer and limb replantation [1]. Despite advancements in microsurgical instruments and techniques, microvascular anastomosis is challenging and requires advanced microsurgical techniques [2,3]. Achieving accurate approximation of anastomosed vessels, eversion of the vessel ends, and proper contact of the intimal layers with no uneven leading points for thrombosis are essential for a successful anastomosis. The conventional technique of microvascular anastomosis with several interrupted sutures is a well-proven method; however, it is still imperfect. Due to the high number of stitches, the technique is time-consuming and increases the ischemia time [4]. Moreover, small blood vessels of 0.3 mm or less at the level of supermicrosurgery do not have a constant location, so it takes a lot of time to find suitable blood vessels in trauma situations [4]. Additionally, a forceps and thin thread suitable for supermicrosugery are required, but if there is no such tool and there is little clinical experience, it takes a lot of time. Also, the stitching procedure causes surgical trauma, and the suture material acts as a foreign body in the lumen, which could lead to thrombosis [5,6]. Therefore, to reduce the surgical time and exposure to suture materials, research on reducing the number of sutures and novel suturing techniques is ongoing [7,8,9,10,11]. Recently, a study showed that anastomosis is possible with three stitches at an angle of 120° [12,13]. In this study, the authors introduced a 2-point suture technique with an angle of 180° to reduce vascular occlusive complications caused by stitches while ensuring lumen eversion. This technique can be used in cases where there is little supermicrosurgery experience or the instrument required for supermicrosurgery is insufficient (e.g., superfine tip forceps or 11-0 or 12-0 sutures). The objective of this study was to evaluate the feasibility, speed, and reliability of this new technique and contribute to ongoing efforts to improve microvascular anastomosis.

## 2. Materials and Methods

### 2.1. Patients Selection and Study Design

From January 2017 to December 2021, the medical records of patients who underwent arterial anastomosis of vessels that were less than 0.3 mm in diameter during replantation at our institution were retrospectively reviewed. Electronic medical records, X-ray films, clinical photographs of the wounds, and demographic information were reviewed. Patient characteristics, preoperative assessments, and operative data (e.g., number of stitches and anastomosis) were recorded. The recorded patient data included age, sex, body mass index, level of the injury, the mechanism of the injury, comorbid medical conditions, anticoagulant use, and tobacco use. We categorized the mechanism of injury into either guillotine (cut) or crush injury. Comorbid medical conditions included hypertension, diabetes, cardiovascular disease, and peripheral artery disease. Postoperative data collected included the anesthesia method, operating time, length of hospital stay, use of hyperbaric oxygen therapy (HBOT), the total number of HBOT sessions, and follow-up duration. All patients included in this study were followed for at least 6 months. Postoperative complications, including arterial insufficiency, infection, tip necrosis, and replantation failure, were defined as complications occurring after anastomosis. We compared the survival rates between those who were treated using the 2-point suture technique and those who were treated with the 3 or 4 sutures.

All study participants provided written informed consent for the storage of their medical information in the database and its use for research purposes. The study protocol was approved by the Institutional Review Board of the Kosin University Gospel Hospital of Korea (KUGH 2022-04-002). All procedures were performed in accordance with the ethical standards of the Institutional and National Research Committee and the 1964 Helsinki Declaration and its later amendments. This study was supported by a National Research Foundation of Korea (NRF) grant funded by the Korean government (MSIT) (No. 2020R1G1A1007678).

### 2.2. Statistical Analysis

A biomedical statistician analyzed the collected data. Statistical analysis was performed using SPSS version 27.0 (IBM Corp., Armonk, NY, USA). Patients’ characteristics were summarized using means with SD or medians with interquartile ranges. Univariate analysis of patient characteristics and complications was conducted using a t-test and Kruskal-Wallis test for continuous variables, Pearson’s chi-square test for categorical variables, and Fisher’s exact test for variables with an expected frequency of less than 5. Odds ratios and 95% confidence intervals were obtained, and a *p*-value less than 0.05 was considered statistically significant.

### 2.3. Operative Procedure and Postoperative Management

Most of the replantations were performed under general anesthesia, and in some cases, nerve block and local anesthesia were performed. First, we found vessels that could be used in wounds and stumps, and vessel preparation for replantation followed standard techniques [14,15]. The 2-point suture technique for anastomosis was performed with 2-points at 180° intervals. A double arm 10-0 Nylon suture (Ethicon, Cornelia, Ga.) was used to pass the thread from the luminal side of the vessel to the outside of the vessel so that the margins were sufficiently everted. The same procedure was performed on the other side, after which a knot was made. Sutures were applied in the same way at the 180° point (Figure 1). We waited for reperfusion, and if blood leakage persisted, an additional suturing was performed in the leakage area. Anastomosis was performed by a senior author (H.-i.K.).

In all the cases, close observation was followed by absolute bed rest for 3 days postoperatively. During monitoring, needle puncture exsanguination was performed if congestion occurred, and revision surgery was performed if arterial insufficiency occurred. If reperfusion was not achieved in the intraoperative field during revision surgery, a tie-over dressing was applied to the amputated part as a composite graft, and HBOT was performed to manage it in the form of a composite graft. By setting 2 atm and 100% oxygen in a hyperbaric chamber, HBOT was performed for 80 min every day for a week. The tie-over dressing for the composite graft was opened after 7 days. All patients were treated with 2 g of intravenous cefazedone sodium twice daily and prostaglandin E (alprostadil alpha-cyclodextrin). Survival was assessed on postoperative day 7, and the patient data, survival rates, and complications were reviewed.

## 3. Results

We enrolled 21 patients (thirteen male and eight female patients) who underwent replantation surgery with arterial anastomosis of vessels with a diameter of less than 0.3 mm between January 2017 and December 2021. The average age of the patients was 44.2 ± 15.1 years (range, 21–68 years). Among the patients, three had diabetes mellitus, three were active smokers, and one was a former smoker. The mechanisms of injury were crush (ten patients) and cut injuries (eleven patients). The injured lesions were on the forehead (one patient), thumb (two patients), index finger (six patients), middle finger (two patients), ring finger (four patients), small finger (five patients), and second toe (one patient). There were ten Tamai level I amputations and ten Tamai level II amputations [16]. In total, fifteen cases of replantation were performed under general anesthesia, three cases were performed under brachial plexus block, and the other three cases were performed under local anesthesia. The mean operating time was 145.2 ± 85.8 min (Table 1).

Of the 21 replantation cases, 19 (90.5%) were successful. Further, the 2-point suture technique was performed in twelve cases, eleven of which survived. Three or four sutures were performed in nine cases, eight of whom survived. The two groups had no statistically significant differences with regard to age, sex, diabetes, or smoking history. Composite graft conversion was performed in three cases where the 2-point suture technique was used, and two of these cases survived. Composite graft conversion was performed in six cases where three or four sutures were used, five of which survived (Table 2). The survival rate was high in the cases where 2-point sutures were used (91.7% vs. 88.9%), and there were fewer cases of conversion to a composite graft (25.0% vs. 66.7%). However, the composite graft conversion rate and survival rate were not statistically significant according to the number of sutures. The average duration of hospital stay was 14.4 days (range: 5–21 days). The average follow-up duration was 13.2 months (range: 12–17 months).

Two patients experienced arterial insufficiency. Wound exploration was performed immediately, and composite graft conversion was performed after re-anastomosis; however, replantation failed. After two weeks of dressing, surgical debridement and full-thickness skin grafting were performed.

Case 1

A 60-year-old woman amputated her left index finger in a plant while using a fish-cutting machine. One digital pulp artery could be found at the site of amputation after debridement of the stump tissue and was anastomosed using the 2-point suture technique. The fingertip survived successfully, and three months postoperatively, the patient did not have any complications or functional impairment (Figure 2).

Case 2

A 61-year-old woman amputated her right index finger in a plant while using an abalone-cutting machine. One digital pulp artery could be found at the site of amputation after debridement of the stump tissue and was anastomosed using the 2-point suture technique. However, the color of the flap became pale in the evening, and arterial insufficiency was noted, so a wound re-exploration was performed, but reperfusion was not achieved. Composite graft conversion was performed after arteriorrhaphy, and it was confirmed that the graft was engrafted seven days after surgery. The fingertip survived successfully, and nine months postoperatively, the patient did not have any complications or functional impairment (Figure 3).

## 4. Discussion

This review suggests that the 2-point suture technique is relatively safe and feasible for microvascular anastomosis. In this study, survival rates were compared between cases where the 2-point suture technique was used and cases where three or four sutures were used. In the 2-point suture group, 11 out of 12 cases survived, and the composite graft conversion rate was 25.0%. In contrast, eight out of nine cases in which three or four sutures were used survived, and the conversion rate was 66.7%. Hence, the outcomes of the 2-point suture technique were satisfactory. We thought that cases with three or four sutures could have poor outcomes due to occlusion of the lumen by the thickness of the suture material and the number of sutures.

The first microscopic surgery was performed in 1921 by Carl Nylen, an otolaryngologist in Stockholm, Sweden [17]. In 1960, Jacobson used a microscope for the first time in vascular surgery for carotid anastomosis in dogs [18]. Since then, further development has continued, and improvements in microsurgical tools, suture materials, vascular clamps, and microscopes have made supermicrosurgery possible [19,20,21]. Koshima et al. published the first use of a perforator flap, the deep inferior epigastric skin flap, in 1989, which heralded the development of a variety of different perforator flaps in the following years [22]. Koshima reported the first use of flaps based on perforator vessels with a caliber of less than 0.8 mm at the First International Course on Perforator Flap and Arterialized Skin Flaps in 1997. The ability to use such small vessels for anastomosis was significant because it greatly increased the surgeon’s freedom in selecting free tissue flaps while at the same time reducing donor site morbidity by preserving fascia, muscles, nerves, and major vessels during the dissection. Koshima et al. first called this technique “supramicrosurgery” in their description of the paraumbilical perforator flap in 1998 [23]. In 2007, Koshima again referred to the technique as “supermicrosurgery” at the first international meeting on innovative microsurgical technology and published a definition of the procedure in 2010 [24]. A consensus on the name “supermicrosurgery” was reached at the First European Conference on Supermicrosurgery held in Barcelona in March 2010 [25]. Supermicrosurgery has many advantages, but its biggest advantage is that it can reduce surgical time and donor site morbidity [25]. Supermicrosurgery is the latest trend in reconstructive surgery and has enabled new flap designs and free tissue transfers, as well as lymphovenous anastomosis [24,25]. However, supermicrosurgery requires considerable technique, as well as specialized instruments and microscopes [25]. In particular, supermicrosurgery requires a higher skill level of eye–microscope–hand coordination, more dexterous tissue handling, and more refined motor skills than microsurgery [26]. A number of training methods have been developed to teach these skills. Among these, the Chen et al. chicken thigh model allows trainees to learn the skills in a comfortable way and to become familiar with the instrument [26]. In the chicken thigh model, the branch of the ischiatic artery and vein is 0.3–0.5 mm, which is optimized for supermicrosurgical practice [26].

Various suture techniques have also been developed, including the simple interrupted microvascular suture, the 12 o’clock to 6 o’clock method, the 3 o’clock to 9 o’clock-side-side method, the triangulation method, and the posterior wall first [1,4,27]. The first description of the 12 o’clock to 6 o’clock microvascular surgery technique is difficult to find in the literature. This technique is considered to be the most basic and is most often referred to as the conventional method of performing a simple interrupted microvascular anastomosis [28]. In this method, the first suture is placed at 12 o’clock (also sometimes referred to as 0 degrees), and the second stitch is placed at 6 o’clock (or 180 degrees). The third and fourth sutures are placed at 2 o’clock and 4 o’clock to complete the anterior wall. The vessel is then turned over 180 degrees with a clamp, and the fifth and sixth sutures are placed at 8 o’clock and 10 o’clock [28]. The main disadvantage of this technique is the need to rotate the vessel 180 degrees to suture the posterior wall, which can potentially cause blood vessel damage. In 1986, Yu et al. presented a method to perform the first suture on the posterior wall and the second suture at a 90 degree angle to the anterior wall to facilitate the remaining sutures and named it the 3 o’clock to 9 o’clock-side-side method [29]. Perform the third, fourth, and fifth sutures on the anterior wall, turn 90 degrees, and perform the remaining three sutures on the posterior wall. According to the authors, the 90-degree rotation reduces potential damage to the endothelium compared to the traditional 180-degree rotation and is easier than suturing the posterior wall first [29]. The triangulation method was first developed by Alexis Carrel in 1902 [30]. It is a method in which three standard stitches are made at an angle of 120 degrees, and then two more stitches are made between each stitch. The lumen can be lifted with two standard stitches, reducing the number of cases where the posterior wall is sutured together [30]. The posterior wall first technique was first described by Harris et al. in 1981 [31]. This technique is also sometimes called the “backup” technique. The first suture is placed in the center of the posterior wall, with the second and third sutures placed on either side of the first. The fourth and fifth sutures are placed adjacent to the second and third, advancing anteriorly on either side, leaving a long tail to facilitate the placement of the sixth and seventh sutures. Similarly, both the sixth and seventh sutures are left long to facilitate the placement of the last suture, which is placed equidistant from the sixth and the seventh. Advantages include constant visualization of the back wall, which reduces the risk of accidentally catching the back wall. Harris et al. concluded that the posterior wall first technique is less complicated, faster, and easier to perform than the anterior wall technique [31].

Amputation is one of the most common cases in emergency departments. Particularly in the case of distal fingertip amputations, microsurgical vascular anastomosis may not be possible, and composite grafts are often used in such nonreplantable fingertip amputations [32]. However, some studies have shown that these composite grafts have good results in treating pediatric fingertip amputations but have a low success rate in adults [33,34]. To increase the success rate of composite grafts in adults, Chen et al. chose to increase the contact area and reduce unnecessary graft thickness [32]. Excision of the bone fragment and defatting of the pulp fat pad on the distal amputated fingertip reduced the thickness of the graft. Circumferential deepithelialization of the amputated stump provided maximum contact surface between the distal amputated fingertip and the wound base. Tie-over wet gauze dressing and finger splinting immobilized the injured fingertip and minimized the risk of a postoperative wound base hematoma and incidental detachment of the graft. These procedures helped adult patients achieve an overall graft survival rate of 93.5% for fingertip amputation [32]. The authors also used the above method when graft conversion was required, according to this study. There were nine cases of graft conversion, seven of which were successful. Composite grafts can be a good alternative for nonreplantable fingertip amputations.

HBOT is a treatment modality in which the patient inhales 100% O_2_ at high atmospheric pressure, and it is well known to have a positive effect on wound healing [35]. Systemic HBOT increases oxygen diffusion in the vessels to improve the condition of ischemia-reperfusion injuries and stimulate angiogenesis [35]. HBOT in plastic surgery is used for wounds, burns, crush injuries, infection, and flap surgery. Previous animal studies have shown that the application of HBOT to composite grafts in rats and rabbits increases graft survival [36,37]. Fordor et al. used 20 Sprague-Dawley rats to harvest 3 × 3 cm skin from the back, sutured the skin to the fascia, and performed a composite graft. Further, by setting 202 kPa and 100% oxygen in a hyperbaric chamber, HBOT was performed for 90 min every day for 2 weeks, and the survival area was larger than that of the control group [36]. Li et al. conducted a rabbit experiment using an auricular composite graft, which is useful for skin defects [37]. They used 24 New Zealand White rabbits to harvest 5 mm, 1 cm, and 2 cm sized circular chondrocutaneous composite grafts from the auricle, and the grafts were sutured to the back. Moreover, by setting 2.4 atm and 100% oxygen in a hyperbaric chamber, HBOT was performed for 90 min, a total of 7 times in 5 days, and the graft survival rate was higher than that of the control group in larger composite grafts [37]. Lee et al. reported that HBOT increased the composite graft survival rate and shortened the graft-healing period in patients with fingertip amputation [38]. HBOT includes intermittent administration of 100% oxygen at pressures >1 atm in a pressure vessel [38]. The arterial PO2 increased to 1000–1500 mmHg owing to dissolved oxygen in the plasma. At tissue and cellular levels, hyperoxygenation promotes angiogenesis and improves post-ischemic tissue survival. Increasing the applied pressure increases the PO2 of tissues, which is beneficial for wound healing [39,40]. The authors performed composite graft conversion in cases of replantation failure and applied 2 atm HBOT for 80 min for a week. All patients tolerated it well and achieved good results.

A limitation of this study was the small sample size and the use of Nylon 10-0 sutures. Due to the small sample size, influencing factors such as age, comorbidities, level of injury, and mechanism of injury may not have been properly assessed. However, the authors’ study did not show a statistically significant difference, and when more samples are collected, the factors associated with the prognosis of anastomosed vessels will be studied. Sutures can also act as foreign bodies or obstacles; therefore, if thinner threads (Nylon 11-0 or smaller sutures) were used, the outcomes of using three or four sutures may have improved [4]. Nowadays, with supermicrosurgical tools, the authors also use 11-0 Nylon, a superfine tip forceps, and perform a lymphovenous anastomosis. In this situation, four to six sutures are performed. Even in the case of microvascular anastomosis at the proximal site, sutures are performed as much as possible to maintain the lumen. However, this technique is an option that can be used in an emergency situation when proper instruments are not available. This study has confirmed that better results can be achieved by attempting the 2-point suture technique rather than many sutures or abandoning vascular sutures due to lack of an instrument.

## 5. Conclusions

The results of this study suggest that the 2-point suture technique is a feasible alternative to placing three or four sutures in microvascular anastomosis. This option may be useful in situations where supermicrosurgical tools are not available. Composite graft conversion and HBOT can be the second choice for reconstruction when replantation failure is suspected.

## Figures and Tables

**Figure 1 jcm-12-02891-f001:**
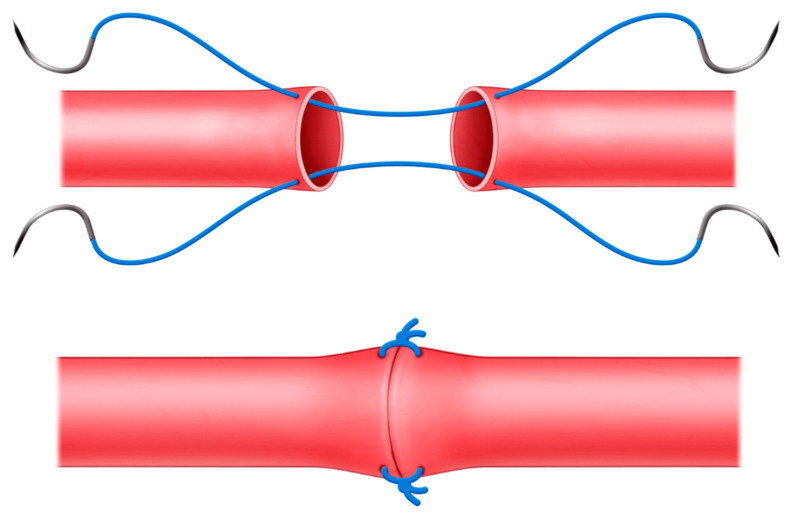
Schematic illustration of the 2-point suture technique.

**Figure 2 jcm-12-02891-f002:**
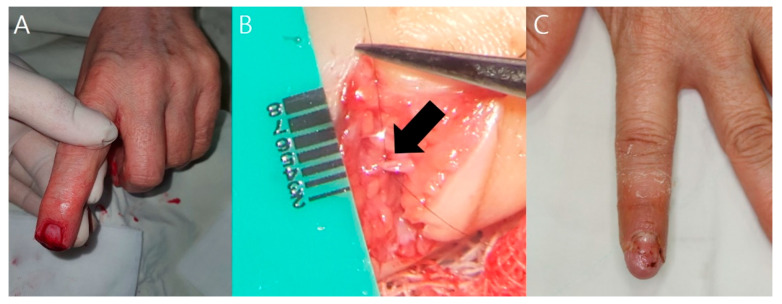
Case 1 (patient 20). (**A**) A 60-year-old patient with an amputated left index finger. (**B**) Black arrow indicates the vessel that was anastomosed using the 2-point suture technique. (**C**) Three months postoperatively, the patient did not complain of any functional complications.

**Figure 3 jcm-12-02891-f003:**
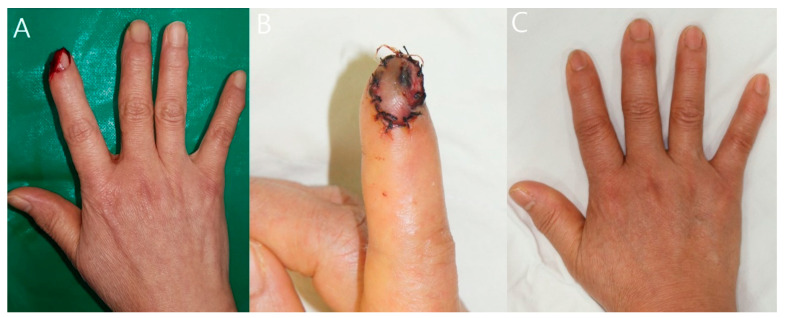
Case 2 (patient 12). (**A**) A 61-year-old patient with an amputated right index finger. (**B**) The tie-over dressing was removed on postoperative day 7. The stump was well maintained. (**C**) Nine months postoperatively, the patient did not complain of any functional complications.

**Table 1 jcm-12-02891-t001:** Summarized patient and operative data *.

	Overall	2-Point Suture Group	3 or 4 Suture Group	*p*
No. of patients	21	12	9	
Survived	19 (90.5%)	11 (91.7%)	8 (88.9%)	1.0
Failed	2 (9.5%)	1 (8.3%)	1 (11.1%)	
Mean age, yrs	44.2 ± 15.1 (21–68)	43.6 ± 17.6 (21–68)	45.1 ± 10.8 (26–62)	0.83
Comorbidities				
Diabetes mellitus	3 (14.3%)	1 (8.3%)	2 (22.2%)	0.55
Active smoker	3 (14.3%)	2 (16.7%)	1 (11.1%)	1.0
Former smoker	1 (4.8%)	1 (8.3%)	0	1.0
Injury type				
Crush	9 (42.9%)	4 (33.3%)	5 (55.6%)	0.40
Cut	12 (57.1%)	8 (66.7%)	4 (44.4%)	0.40
Anesthesia method				
General	15 (71.4%)	9 (75%)	6 (66.7%)	1.0
Nerve block	3 (14.3%)	2 (16.7%)	1 (11.1%)	1.0
Local	3 (14.3%)	1 (8.3%)	2 (22.2%)	0.55
Mean operating time, min	145.2 ± 85.8 (50–360)	152.9 ± 82.4 (50–345)	135.0 ± 89.1 (50–360)	0.66
Replantation part				
Forehead	1 (4.8%)	0	1 (11.1%)	0.43
Finger	19 (90.5%)	12 (100%)	7 (77.8%)	0.17
Toe	1 (4.8%)	0	1 (11.1%)	0.43
Amputated level ^†^				
Zone I	10 (47.6%)	6 (50%)	4 (44.4%)	1.0
Zone II	10 (47.6%)	6 (50%)	4 (44.4%)	1.0
No. Composite graft conversion and HBOT	9 (42.9%)	3 (25%)	6 (66.7%)	0.09
Mean hospital days	14.0 ± 4.4	13.8 ± 4.6	14.3 ± 4.1	0.78
Mean follow-up period, months	13.2 ± 1.8	13.1 ± 2.1	13.4 ± 1.2	0.67

* Values are expressed as median (interquartile range), n (%), mean ± SD, or mean (range). ^†^ Tamai classification for fingertip amputation.

**Table 2 jcm-12-02891-t002:** Patients’ baseline characteristics and operative data.

Cases	Age (Years)/Sex	Mechanism of Injury	Injured Lesion	Anastomosis Stitches	Composite Graft and HBOT	Outcome
1	26/M	Crush	Forehead	3	X	Survived
2	52/F	Cut	LSF Zone I *	2	X	Survived
3	56/M	Crush	RSF Zone II *	4	O	Failed
4	49/M	Cut	RRF Zone II *	3	O	Survived
5	21/M	Crush	RMF Zone I *	2	X	Survived
6	51/F	Crush	LRF Zone I *	3	O	Survived
7	65/M	Cut	LIF Zone II *	2	X	Survived
8	46/M	Cut	LIF Zone II *	3	X	Survived
9	26/F	Cut	LMF Zone II *	2	X	Survived
10	32/M	Crush	RT Zone II *	2	X	Survived
11	53/F	Cut	RT Zone I *	2	X	Survived
12	61/F	Cut	RIF Zone I *	2	O	Survived
13	45/M	Cut	RIF Zone I *	4	O	Survived
14	31/F	Cut	RSF Zone I *	4	O	Survived
15	21/M	Cut	RSF Zone II *	2	X	Survived
16	62/M	Crush	LSF Zone II *	3	O	Survived
17	68/M	Crush	RIF Zone I *	2	O	Survived
18	22/M	Cut	LRF Zone II *	2	X	Survived
19	42/M	Crush	RRF Zone II *	2	O	Failed
20	60/F	Cut	LIF Zone I *	2	X	Survived
21	40/F	Crush	L 2nd toe Zone I *	3	X	Survived

M, male; F, female; HBOT, hyperbaric oxygen therapy. RT, right thumb; RIF, right index finger; RMF, right middle finger; RRF, right ring finger; RSF, right small finger. LIF, left index finger; LMF, left middle finger; LRF, left ring finger; LSF, left small finger. * Tamai classification for fingertip amputation.

## Data Availability

Not applicable.

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
