# Peer review of "What Is the Minimum Number of Sutures for Microvascular Anastomosis during Replantation?"

_jcm, 2023, doi:10.3390/jcm12082891_

Round 1

Reviewer 1 Report

I congratulate the authors for their hard work and innovative idea, which deserves attention and further attention and possible wide implementation. 

I have a few comments though.

1. introduction

- are nine sutures a magic number? or the anastomosis requires normally nine sutures but each clinical situation lends to the individual solution (10,11,12 sutures). Also, the diameter of the vessel should be taken into account.

- does a routine microsurgical anastomosis is time-consuming? In my experience, 10-15 minutes are sufficient for an arterial anastomosis for 1-3 mm vessels with the triangulation technique. So it does not increase ischemia time. 

- the authors have introduced a new technique for small vessels (under 0.3 mm in diameter - super microsurgery) on what basis - does they have laboratory experience or any other research on this matter (citations?)

2. M&M

- on what basis were the patients where assigned to 2 - point or 3 - point sutures? it is not clear to me. Or the cases that needed more than 2-point where assigned to the 3-point group?

4. Discussions 

- There is a lot of debate if Nylen, Carrel or Jacobson was the first who has used a microscope in surgery. 

- I would enjoy a more detailed comparison between the technique proposed by the authors, the 3-point technique, and other options available for supermicrosurgery 

- the authors highlight the study limitations - how could they address this in the future? any new directions for research? They are stating that the use of Nylon 10-0 sutures is a limitation. Indeed, when dealing with under 0.3 mm vessels the 10-0 sutures may be not small enough - did they consider using 11-0 sutures (or smaller) - in this case maybe there was "room" enough for 3, or 4 sutures? I would like to hear their opinion on this matter.  

Do they believe that a 2-point suture is more hemodynamically favorable with 10-0 vs., let's say a 3, or 4-point suture with a smaller suture, like 11-0?  

Overall, I do believe that the article brings a new technique to the discussion, that could optimize the very small vessel anastomosis, in order to gain time and patency rates.  

Author Response

Point-by-Point Responses to the Reviewer's Comments

We thank you for your time and consideration of our submission. Below, we address the reviewer’s comments, including changes that we made to our manuscript according to their reports. We believe that these modifications have strengthened the manuscript and hope that the revised manuscript is suitable for publication in Journal of Clinical Medicine.

Point 1: Are nine sutures a magic number? or the anastomosis requires normally nine sutures but each clinical situation lends to the individual solution (10,11,12 sutures). Also, the diameter of the vessel should be taken into account.

Response 1: Thank you for your time and consideration of our submission. There is no standard number, but nine sutures were mentioned as an example, and it is correct to determine sutures considering the thickness of the thread and the diameter of the vessel. The content has been changed to avoid misunderstanding in introduction section line 32.

Point 2: Does a routine microsurgical anastomosis is time-consuming? In my experience, 10-15 minutes are sufficient for an arterial anastomosis for 1-3 mm vessels with the triangulation technique. So it does not increase ischemia time.

Response 2: Thank you for the good point out. It may not take too much time to suture a large blood vessel (>1mm), but what the authors want to say is that small blood vessels of 0.3 mm or less at the level of supermicrosurgery do not have a constant location, so it takes a lot of time to find blood vessels in trauma situations. In addition, a forcep and thin thread suitable for supermicrosugery are required, but if there is no such tool and there is little clinical experience, it takes a lot of time.

What we trying to say is to introduce techniques that can be used in cases where there is little supermicrosurgery experience or no tools are available. In accordance with the reviewer’s opinion, the average operative time was added to Table 1.

Point 3: The authors have introduced a new technique for small vessels (under 0.3 mm in diameter - super microsurgery) on what basis - does they have laboratory experience or any other research on this matter (citations?)

Response 3: In Foo's study mentioned in line 42 of the Introduction, 3-stitch suturing was attempted at an angle of 120 degrees, and in Cifuentes' study, it was confirmed through animal experiments that this method was suitable for supermicro sugical skill training [1,2]. Based on this, the authors wondered what the minimum number of stitches should be for a microvascular anastomosis of 0.3 mm or less, and performed the anastomosis by first using two stitches, confirming that the lumen was maintained, and then adding additional stitches. As pointed out by the reviewer, reference to animal experiment was added in introduction line 42.

  1. Foo TL. Open guide suture technique for distal fingertip replantation. J Plast Reconstr Aesthet Surg. 2013;66:443–4.
  2. Cifuentes IJ, Rodriguez JR, Yanes RA, et al. A Novel Ex Vivo Training Model for Acquiring Supermicrosurgical Skills Using a Chicken Leg. J Reconstr Microsug. 2016;32:699-705.

Point 4: On what basis were the patients where assigned to 2 - point or 3 - point sutures? it is not clear to me. Or the cases that needed more than 2-point where assigned to the 3-point group?

Response 4: It was not classified according to the patient case, but determined according to the process in the situation of microvascular anastomosis of 0.3 mm or less. 2-point suture was made and waited for after reperfusion, and if blood leakage persisted, an additional suture was made in the leakage area. In most cases where more than 2 stitches were performed were 3 stitches and occasionally 4 stitches. The authors also struggled with how to present this. To reduce ambiguity, we will present the number of stitches for each patient in Table 2 and method section line 98.

Point 5: There is a lot of debate if Nylen, Carrel or Jacobson was the first who has used a microscope in surgery.

Response 5: The first microscopic surgery was performed in 1921 by Carl Nylen, an otolaryngologist in Stockholm, Sweden. However, Jacobson was the first to use a microscope for vascular surgery in the 1960s. We added that to the discussion line 194 to clear up any misunderstandings.

Point 6: I would enjoy a more detailed comparison between the technique proposed by the authors, the 3-point technique, and other options available for supermicrosurgery.

Response 6: We are introducing techniques that can be used in situations where there is no superfine tip forceps or 11-0 nylon required for supermicrosurgery. This is an option that can be used in an emergency situation when equipment is not available. In the case of elective surgery, the authors also use 11-0 Nylon, a superfine tip forcep and perform a lymphovenous anastomosis. Vascular sutures using fine tools and 11-0 threads have better results than 2-point sutures performed with 10-0 threads, However, if you do not have fine tools and 11-0 thread, it will be helpful to use the 2-point suture technique rather than many sutures.

Point 7: The authors highlight the study limitations - how could they address this in the future? any new directions for research? They are stating that the use of Nylon 10-0 sutures is a limitation. Indeed, when dealing with under 0.3 mm vessels the 10-0 sutures may be not small enough - did they consider using 11-0 sutures (or smaller) - in this case maybe there was "room" enough for 3, or 4 sutures? I would like to hear their opinion on this matter.

Response 7: That's a good point. 10-0 Nylon is 0.02mm thick and 11-0 Nylon is 0.01mm thick, so with 11-0 Nylon and a superfine tip forceps, 4 stitches or more are possible. Our topic in this study is to introduce an option that can be useful in situations when the necessary equipment for supermicrosurgery is not available. Therefore, this study has confirmed that better results can be achieved by attempting at least a 2-point suture, rather than abandoning vascular suture due to lack of equipment. We wrote this article because we wanted to inform subscribers of the importance of continuing to the challenge and research on microsurgery, even if they are in a harsh environment. We will also add about this in the discussion as well.

Point 8: Do they believe that a 2-point suture is more hemodynamically favorable with 10-0 vs., let's say a 3, or 4-point suture with a smaller suture, like 11-0?

Response 8: As the diameter of 11-0 Nylon is half that of 10-0, the area is 1/4, so numerically the 4-point suture with 11-0 blocks half the area of the lumen rather than the 2-point suture with 10-0. Therefore, it is correct to suture with 11-0 Nylon, but we would like to introduce an option that can be done in a situation where the equipment required for supermicrosurgery is insufficient. We will also add this to the discussion line 304.

Point 9: Overall, I do believe that the article brings a new technique to the discussion, that could optimize the very small vessel anastomosis, in order to gain time and patency rates.

Response 9: Thanks for the reviewer's comments. We look forward to sharing these results with our readers.

Reviewer 2 Report

I would like to congratulate the authors on their well written manuscript and the impressive results. However, there are a few points that the authors should address:

In the introduction section, line 34, it is statet that the traditional nine point suture has worse outcomes. Please provide literature for this statement.

How many stitches were performed in the group with >2 stitches? 

Personally, I feel that the forehead and toe case should be excluded from the study to improve the focus of the manuscript.

Please also add the operating times.

The discussion section is rather short and superficial. Is there data on the 2-point technique in lympatic surgery? Please discuss the advantages and disadvantages of the used techniques in more detail. Is there more literature on the application of HBOT in replantation? Please elucidate further. How are your experiences in the replantation of more proximal amputations?

Author Response

Point-by-Point Responses to the Reviewer's Comments

We thank you for your time and consideration of our submission. Below, we address the reviewer’s comments, including changes that we made to our manuscript according to their reports. We believe that these modifications have strengthened the manuscript and hope that the revised manuscript is suitable for publication in Journal of Clinical Medicine.

Point 1: In the introduction section, line 34, it is statet that the traditional nine point suture has worse outcomes. Please provide literature for this statement.

Response 1: Thank you for your time and consideration of our submission. What the authors were trying to say is that the more suture knots there are, the more they can act as a foreign body or obstacle cause thrombosis [1,2]. We added the references to this as 5,6 and misunderstandings were corrected in the introduction section line 33.

  1. Akentieva TN, Ovcharenko EA, Kudryavtseva YA. Influence of suture material on the development of postoperative complications in vascular surgery and their prevention. Khirurgiia. 2019;10:75-81.
  2. Dahlke H, Dociu N, Thurau K. Thrombogenicity of different suture materials as revealed by scanning electron microscopy. J Biomed Mater Res. 1980;14:251-68.

Point 2: How many stitches were performed in the group with >2 stitches?

Response 2: Thank you for the good point out. 2-point suture was made and waited for after reperfusion, and if blood leakage persisted, an additional suture was made in the leakage area. In most cases where more than 2 stitches were performed were 3 stitches and occasionally 4 stitches. The authors also struggled with how to present this. To reduce ambiguity, we will present the number of stitches for each patient in Table 2 and method section line 98.

Point 3: Personally, I feel that the forehead and toe case should be excluded from the study to improve the focus of the manuscript.

Response 3: We respect the reviewer’s opinion, but have added a few cases to let subscribers know that this technique can be applied to other amputation or defect as well as fingertips.

Point 4: Please also add the operating times.

Response 4: According to the reviewer's advice, operating time was added to Table 1.

Point 5: The discussion section is rather short and superficial. Is there data on the 2-point technique in lympatic surgery? Please discuss the advantages and disadvantages of the used techniques in more detail. Is there more literature on the application of HBOT in replantation? Please elucidate further. How are your experiences in the replantation of more proximal amputations?

Response 5: Thank you for the good points that can improve the quality of the journal. As pointed out by the reviewers, we have added content throughout the journal. In particular, the discussion added the thoughts of the authors on the results, the relationship between HBOT and replantation, composite graft, suture methods, and supermicrosurgery. Also we added a case presentation in result section line 153. At the proximal site, more sutures are performed because sufficient lumen can be secured. Contents of proximal site anastomosis was also added to the discussion.